# Association between Branched-Chain Amino Acid Intake and Physical Function among Chinese Community-Dwelling Elderly Residents

**DOI:** 10.3390/nu14204367

**Published:** 2022-10-18

**Authors:** Minqi Liao, Yingjun Mu, Xin Su, Lu Zheng, Shiwen Zhang, Hongen Chen, Shan Xu, Junrong Ma, Ruiqing Ouyang, Wanlin Li, Chen Cheng, Jun Cai, Yuming Chen, Changyi Wang, Fangfang Zeng

**Affiliations:** 1Department of Non-Communicable Disease Prevention and Control, Shenzhen Nanshan Center for Chronic Disease Control, Shenzhen 518000, China; 2Department of Public Health and Preventive Medicine, School of Medicine, Jinan University, No. 601 Huangpu Road West, Guangzhou 510630, China; 3Institute of Epidemiology, Helmholtz Munich-German Research Center for Environmental Health, D-85764 Munich, Germany; 4Disease Control and Prevention Institute, Jinan University, Guangzhou 510630, China; 5Jinan University-BioKangtai Vaccine Institute, School of Medicine, Jinan University, Guangzhou 510630, China; 6Guangdong Provincial Key Laboratory of Food, Nutrition and Health, Department of Epidemiology, School of Public Health, Sun Yat-sen University, Guangzhou 510080, China

**Keywords:** branched-chain amino acids, elderly population, muscle strength, physical function

## Abstract

This study aimed to evaluate the potential associations of dietary BCAAs (isoleucine, leucine, and valine) with physical function in the elderly Chinese population. A validated semiquantitative food frequency questionnaire and anthropometric and physical function measurements were used to collect data. We modeled trends in physical function indicators for BCAA quartiles using multivariate linear regression models. Among 4336 (43.97% men) participants aged 72.73 ± 5.48 years, a higher dietary intake of BCAAs was positively associated with increased handgrip strength (all *p* trends < 0.001), shorter times for 4-m fast walking (all *p* trends < 0.001) and repeated chair rises (all *p* trends < 0.001). No linear association was found between subtypes of amino acids and any physical functions (all *p* trends > 0.05). Individuals in the highest quartiles of BCAA intake had a reduced risk of developing low muscle strength, and the multiadjusted odds ratios (ORs) and 95% confidence intervals (95% CIs) for women and men were 0.50 (0.38–0.65) and 0.67 (0.50–0.91), respectively. Similarly, higher BCAA consumption was associated with a lower risk of developing low physical performance (4-m walking speed: OR = 0.68 [0.50–0.93]; repeated chair rises: OR = 0.66 [0.54–0.81]). Higher dietary BCAA intake might be beneficial for physical function in the elderly population.

## 1. Introduction

Approximately 25% of the population in Asia is projected to be ≥60 years old by 2025 [1]. Aging is an irreversible process that is one of the most important risk factors for functional decline [2]. One of the manifestations of aging is a decrease in motor capacity due to the loss of skeletal muscle mass, strength, and function [3]. A decline in physical function could impair the ability to live independently, reduce the quality of life among the older population, increase mortality and place a huge burden on the health and social care systems [4]. In this respect, the identification of core strategies for the management and prevention of physical performance is of great value in maintaining a fulfilling life among older adults.

Although the loss of muscle power and strength could be accelerated with aging, there is wide interindividual variability in declines in muscle strength and mass among elderly individuals because of potentially modifiable risk factors, including exercise, diet, and nutrition [5]. In particular, an insufficient dietary intake of protein could speed up the declines in muscle function, bone strength, and immunity response among older adults [6]. Similarly, a protein intake above 0.8 g protein/kg/day could exert a protective effect against hip fractures [7] and bone mass density loss and be beneficial to maintaining physical function [8,9,10,11]. Proteins are complex mixtures of amino acids that can exert different effects on muscle protein synthesis and physical performance.

Branched-chain amino acids (BCAAs), including leucine, isoleucine, and valine, are the most abundant essential amino acids in proteins but can only be synthesized in bacteria, plants, and fungi rather than in animals [12]. This means that diet, rather than endogenous synthesis, is the major significant source of BCAAs for humans [12]. BCAAs are known to maintain protein synthesis, energy production, and the synthesis of many neurotransmitters [13]. Numerous reports have linked BCAAs with age-related loss of physical function, commonly measured by handgrip strength, gait speeds, and repeated chair rises [14]. For instance, a cross-sectional study involving 227 older adults found that lower levels of BCAAs in serum were associated with a lower skeletal muscle index (SMI), handgrip strength, and longer repeated chair rise time (five times) in community-dwelling older adults [15]. A series of randomized controlled trials (RCTs) have further demonstrated the protective effect of dietary BCAAs against decreased physical performance [16,17,18]. However, these RCTs were mainly performed among functionally limited elderly patients who might be more sensitive to nutritional intervention than healthy free-living older individuals.

Considering that a limited number of previous studies have explored associations between BCAAs and physical function in elderly populations, with their findings greatly limited by their small sample sizes, most of the prior studies rarely examined the effects of BCAAs as a whole or explored the effects of specific BCAA types on muscle strength and physical function. Hence, the present study aimed to explore the relationship between dietary intake of BCAAs and physical performance in an elderly community-based population by using handgrip strength, 4-m walking speeds (usual and fast walking speeds), and repeated chair rise as the main outcomes. Understanding the effects of dietary BCAAs on physical performance may provide new opportunities to improve health and independent living in the elderly population.

## 2. Materials and Methods

### 2.1. Participants

Data for this cross-sectional study were extracted from the Nanshan elderly population cohort, which was established based on the National Free Health Examination Project in community-dwelling elderly residents according to China’s national health policy and aimed to investigate the nutritional and health status of Chinese adults aged 65 years and older. By using a stratified cluster random sampling method, this project enrolled participants from 53 community health service centers in eight blocks in the Nanshan District, Shenzhen, China, between May 2018 and December 2019. A total of 4478 potential participants were initially recruited, but we only included eligible participants who (i) were aged 65 years old and above; (ii) had lived in Shenzhen for at least 6 months; (iii) had undergone annual physical examinations at community health service centers; and (iv) agreed to participate and sign the informed consent form. Participants were excluded if they met any of the following criteria: (i) age under 65 years (*n* = 3); (ii) declined to participate or reported difficulties in routine communication or activities (*n* = 6); (iii) reported implausibly low or high dietary energy intake (<600 kcal/day (*n* = 77) or >4000 kcal/day (*n* = 43)); and (iv) did not undergo the annual physical examination (*n* = 19).

Ultimately, a total of 4336 qualified participants were identified in the analysis. Written consent was obtained from all of the participants, and the study protocol was approved by the Ethics Committee of the Shenzhen Nanshan Center for Chronic Disease Control (No. 1120180009).

### 2.2. Covariate Collection

Potential confounders such as sociodemographic factors (age, sex, household registration, body mass index [BMI, kg/m^2^], etc.), health-related behaviors (smoking status, alcohol consumption, and physical activities), medical histories (diabetes, hypertension, dyslipidemia, etc.), and family histories of diseases were collected by trained investigators with relevant medical knowledge through face-to-face interviews using a structured questionnaire. The latest laboratory data including fasting blood glucose (GLU), total cholesterol (TC), total triglycerides (TG), high- and low-density lipoprotein (HDL-C and LDL-C) were collected from electronic reports in community health service centers. Weight and height were measured, and BMI was calculated as weight (kg)/height (m^2^).

### 2.3. Dietary Assessment

Habitual dietary consumption was assessed with a tool derived from a validated food frequency questionnaire (FFQ) [19] which was conducted based on food intake during the month preceding the interview. With some uncommonly consumed foods not included due to differences in dietary habits, 62 food items were organized based on a validated semiquantitative 81-item FFQ, which has previously been verified by six 3-day energy-adjusted diet records of 26 nutrients among women in Guangzhou [19]. With a common unit or portion size specified for each food item (in bowls, boxes, cups, grams, etc.), participants were asked to report their average food consumption in five frequencies (never, yearly, monthly, weekly, and daily). Colored pictures of foods in corresponding portion sizes were provided to help quantify the food portions. Each food consumption was converted into the daily intake (g/d), and daily average energy and nutrient intakes were estimated using the Chinese Food Composition Table, 2009 [20]. Intakes of total BCAAs were calculated as the cumulative sum of the three amino acids (leucine, isoleucine, and valine).

### 2.4. Muscle Strength and Functional Performance Measures

Major physical function parameters in the present study included handgrip strength, 4-m walking speeds (usual and fast walking speeds), and repeated chair rises. The handgrip strength (in kilograms) was measured by asking participants to stand in an upright position and to hold the dynamometer in the dominant hand with the arm unsupported and parallel to the body [21]. Handgrip strength was measured three times, and the highest measurement of handgrip strength was registered. The usual and fastest walking speeds (in m/s) were measured two times over a 4-m course in an unobstructed and dedicated corridor [22]. The timing began when the participants moved from the initial starting point and stopped when they crossed the 4-m mark [23]. The faster of the two gait speeds was used in this analysis. The repeated chair rises (in seconds) were measured by asking the participants to stand up from a chair with their hands folded across their chest five times as quickly as possible, with a digital stopwatch used to record the time taken to complete the task [22].

### 2.5. Criteria for Physical Function Decline

Based on the Asian Working Group for Sarcopenia (AWGS) criteria, low muscle strength is defined as handgrip strength < 28 kg and < 18 kg for men and women, respectively; low physical performance is defined as a 4-m walk < 0.8 m/s or repeated chair rises ≥12 s [24,25].

### 2.6. Statistical Analyses

The normal distribution of continuous variables was analyzed using the Kolmogorov–Smirnov test, and log transformation was conducted for continuous variables that were not normally distributed. Continuous variables are presented as the means ± standard deviations (SDs) or medians, unless stated otherwise, whereas categorical variables are described as numbers and percentages. Dietary intake of nutrients was adjusted for total energy intake using the residual method, and the energy-adjusted dietary intake of the BCAAs was divided into quartiles (Q1–Q4).

Mean differences (MDs) and changes in characteristics among subjects according to the BCAA quartiles were detected by using analysis of covariance (ANCOVA) for continuous variables and the chi-square test for categorical variables. Three multivariate linear regression models were developed for the association analysis: based on the crude model that did not adjust for any confounders, Model 1 was adjusted for age and sex; Model 2 was further adjusted for BMI, smoking and drinking status, diabetes, and hypertension [26]; and Model 3 was adjusted for covariates in Model 2 plus soluble fiber intake level [27]. *p* for the linear trend was calculated by treating quartiles as continuous variables. Using unconditional logistic regression analysis, we obtained odds ratios (ORs) and their 95% confidence intervals (95% CIs) in four models to assess the associations of BCAA intake with the risk of physical function decline.

Separate analyses were used to explore the associations of each amino acid (isoleucine, leucine, and valine) with four indicators of physical function. To demonstrate the robustness of our findings, sensitivity analyses were also performed on subpopulations excluding those who had been diagnosed with coronary heart disease, myocardial infarction, stroke, and angina pectoris. All statistical analyses were conducted with R software (version 4.1), with a two-tailed *p* < 0.05 considered statistically significant.

## 3. Results

The characteristics of the participants are shown in Table 1, with the details of selection shown in Figure 1. Of the 4336 participants, 42.97% were men, and the mean age was 72.73 ± 5.48 years. Compared to subjects with the least dietary exposure to BCAAs (Q1: <13,210.90 mg/day), those most exposed (Q4: ≥24,360.62 mg/day) were more likely to be male and to live in large cities, be a current drinker, and not have hypertension (all *p* trends < 0.05). In addition, participants in the highest quartile, on average, had higher levels of handgrip strength but had shorter times for 4-m fast walking and repeated chair rises (all *p* trends < 0.001).

Dietary intakes of seven leading nutrients, including energy intake, fat, protein, carbohydrate, dietary soluble fiber, vitamin D, and folate were calculated in all of the populations of interest. The mean levels of all seven nutrients according to the quartiles of dietary BCAA levels are shown in Table 2. Compared to those with the lowest BCAA intake (Q1), those with the highest dietary intake of BCAAs (Q4) tended to have higher intake levels of energy, but have lower levels of fat and vitamin D (all *p* trends < 0.05), with those with higher BCAA intake (Q3) having the highest intakes of carbohydrate and folate (both *p* trends = 0.005), providing evidence of positive linear relationships between SD increases in BCAA intakes and energy, carbohydrate, and folate intakes. In turn, the intake of protein (*p* trend = 0.201) and dietary soluble fiber (*p* trend = 0.377) did not differ across BCAA quartiles.

Appendix A presents the distribution of seven nutrients across the quartiles of the amino acids isoleucine, leucine, and valine. We observed positive associations of all three amino acids with the intake of carbohydrates and folate (all *p* trends < 0.05) but found inverse associations with dietary consumption of vitamin D (all *p* trends < 0.05). For specific BCAAs, isoleucine and valine intakes were both inversely associated with fat consumption (both *p* trends < 0.05), whereas leucine and isoleucine were positively associated with fat intake (*p* trend = 0.027) and dietary soluble fiber intake (*p* trend = 0.031), respectively. No significant trend was observed in the distribution of energy intake and protein among all three BCAAs (*p* trends ranged from 0.607 to 0.968).

Table 3 displays the results of the crude and multivariable models relating physical performance indicators and total BCAA intake. In the linear regression models with four physical function indicators as the dependent variables, the quartiles of dietary BCAAs showed a linear trend for a positive association between total BCAA intake and increased handgrip strength, regardless of the crude model or multivariate adjustment (Models 1–3, the MD ranged from 1.64 to 2.97; all *p* trends < 0.001). Conversely, a linear trend for an inverse association between dietary BCAA intake and the time for 4-m fast walking persisted in four models (MD ranged from −0.28 to −0.24; all *p* trends < 0.001), while the inverse association between BCAA intake and time for 4-m usual walking changed after adjustments. Repeated chair rises yielded similar results (MD ranged from −1.11 to −0.83; all *p* trends < 0.001), with these associations barely changing after additional adjustments for potential covariates.

Table 4 shows the association between the intake of total BCAAs and a reduced risk of developing weak muscle strength and physical function decline. Higher dietary exposure to BCAAs was associated with a decreased risk of weak muscle strength for quartiles 2–4 of exposure compared to the lowest quartile after full adjustments in women, and their full-adjusted ORs and 95% CIs were 0.72 (0.56–0.92), 0.62 (0.48–0.80), and 0.50 (0.38–0.65), respectively. For men, only the second and fourth quartiles of dietary BCAA intake were inversely associated with the risk of weak muscle strength (full-adjusted ORs: 0.67 [95% CI: 0.49–0.91] and 0.67 [95% CI: 0.50–0.91]). For a decline in physical performance, only dietary BCAA intake in the highest quartile was negatively related to a slow 4-m usual walking speed (full-adjusted OR: 0.68 [95% CI: 0.50–0.93]) when compared to the lowest heading quartile. Meanwhile, those from higher quartiles (Q3 and Q4) of dietary BCAA intake had an almost 33%~44% reduced risk of slow repeat chair rises, and this association did not vary after adjusting for all confounders (full-adjusted OR: 0.77 [95% CI: 0.63–0.94] for Q3 and 0.66 [95% CI: 0.54–0.81] for Q4), while a similar result was found only among participants in Q2 in Models 2 and 3 that further controlled for health-related behaviors and specific nutrients (full-adjusted OR: 0.79 [95% CI: 0.64–0.96]).

Regarding individual amino acids, as shown in Table 5, a null association was observed between all three amino acids and all of the four physical function indicators (all *p* trends > 0.05).

Appendix A show the results of analyses conducted among a subset of populations to control for the potential influence of the major cardiovascular diseases (coronary heart disease, myocardial infarction, stroke, and angina pectoris), and the findings presented only minimal differences in most of the cases. The positive association between total dietary exposure to BCAAs and handgrip strength remained significant in all four subpopulations (all *p* trends < 0.001). Similarly, negative associations of total BCAA intake with a shorter time for the 4-m fast walking time or repeated chair rise time were obtained in all four subpopulations (all *p* trends < 0.001). The inverse association between total BCAA intake and 4-m usual walking time remained robust in the crude model among the subpopulations excluding those with coronary heart disease (crude *p* trend = 0.025) and myocardial infarction (crude *p* trend = 0.018), but this association disappeared in other subpopulations excluding those with stroke (crude *p* trend = 0.494) and angina pectoris (crude *p* trend = 0.484).

## 4. Discussion

In this cross-sectional study, we found that participants with higher exposure to dietary BCAAs had a decreased risk of developing weak muscle strength or physical performance decline compared with those with lower dietary BCAA intake. These findings were robust to the exclusion of participants with cardiovascular diseases. No significant association between individual amino acids and any physical function indicators was observed. We also noticed that a higher dietary BCAA intake was positively associated with the higher dietary consumption of energy, carbohydrate, and folate, but inversely associated with fat and vitamin D intake.

Prior studies have provided supporting evidence for the association of BCAAs with age-related loss in physical performance among older adults. For example, a cross-sectional study conducted on 3292 Korean individuals indicated a significant association between high dietary BCAAs and increased SMI, an indicator of the amount of skeletal muscle mass [28], but this study was performed among middle-aged adults (aged 50–64 years) rather than elderly adults over 65 years of age. Their subsequent study conducted among 4852 Korean older participants (aged ≥ 65 years) also found that an increased dietary BCAA intake, particularly leucine, was positively associated with improved handgrip strength [29]. However, this study mainly focused on handgrip strength, leaving a gap in assessing the potentially beneficial effects of BCAA intake on other related functional indicators.

Moreover, a cross-sectional study conducted among 227 community-dwelling adults aged ≥ 65 years in the Netherlands suggested that lower dietary exposures to BCAAs and leucine were both related to lower levels of SMI and handgrip strength but longer time for repeated chair rises (all *p* < 0.05) [15]. An RCT conducted on 140 European older adults (≥65 years) indicated that four weeks of protein-based supplementation (foods dominant in leucine and vitamin D) significantly increased the 4-m gait speed (0.061 m/s, 95% CI: 0.043–0.080, *p* < 0.001) and muscle mass (*p* < 0.03), improved the rehabilitation intensity profile (*p* = 0.003), and also decreased the rehabilitation (*p* < 0.001) and hospital stay time (*p* < 0.001) [30]. A quasi-experimental study including 33 subjects with presarcopenia or sarcopenia (average age 66.6 ± 10.3 years) further demonstrated that five weeks of BCAA supplementation significantly improved the SMI, gait speed, and grip strength (all *p* values < 0.05) [31].

Aging affects biological functions and increases vulnerability to many diseases by reducing mitochondrial biogenesis, stimulating mitochondrial malfunction, and increasing oxidative damage [32,33]. Previous studies have provided substantial evidence relating dietary BCAAs to multiple mechanisms underlying the pathogenesis of age-related loss in physical function. Long-term dietary supplementation with a specific BCAA-enriched mixture could improve the function of cardiac and skeletal muscles, enhancing physical endurance and motor coordination by increasing mitochondrial biogenesis and upregulating the expression of sirtuin 1, which is related to lifespan extension, enhanced mitochondrial biogenesis, and decreased reactive oxygen species (ROS) production [34]. An oral intake of leucine, but not isoleucine or valine, could effectively stimulate protein synthesis in skeletal muscle cells [35] by stimulating the anabolic signaling mammalian target of rapamycin complex 1 and other factors related to protein synthesis [36], thus facilitating muscle protein synthesis rates [37]. Furthermore, BCAA intake improves muscle function by decreasing the concentration of creatine kinase, a marker of muscle damage [38].

We found a positive association between BCAA intake and improved fast walking speed [39] in contrast to previous studies relating BCAAs to improved usual gait speed [40,41]. This association was observed only in the crude model in the present study, and it might be partly explained by the effects of potential confounders in our study. Furthermore, compared to the usual gait speed, a marker of daily activity with lower frequency and intensity of exercise, fast walking speed is a better predictor of functional decline [42] because it is highly dependent on muscle strength and lower extremity functioning. A combination of walking for exercise more frequently and amino acid intake could improve the absorption response and musculoskeletal sensitivity to amino acids, thus enhancing muscle protein synthesis [43].

Regarding specific amino acids, a null association of all three BCAAs (isoleucine, leucine, and valine) with all four physical function indicators was found in separate analyses. However, leucine improves physical function in the elderly population by promoting the synthesis of skeletal muscle [26]. A low serum concentration of leucine was associated with decreased muscle mass, strength, and function in older adults [15]. A cross-sectional study reported an independently positive association between dietary leucine intake and increased muscle mass and strength among healthy older individuals in Brazil [44]. These contradictory results may be partly ascribed to the interactions among different BCAAs, and the amount of each of the three amino acids specifically did not reach the effective value after separating the influence of any one amino acid from the others [45]. In this regard, further prospective research is needed to determine if these findings persist in a larger sample size.

For the association of BCAA intake and seven main nutrients, more dietary BCAA intake means higher consumption of foods containing different protein sources, including animal proteins and protein-rich plant foods such as soybeans or cereals, which also contain high amounts of carbohydrates and/or fat, thus increasing the risk of excessive total energy intake [46]. Furthermore, as a good source of amino acids, meat is also the main source of B vitamins, which could partly explain the positive relationship between the intake of BCAAs and folate. In addition, lower intakes of fat might be related to the recommendation to consume a healthy high-protein low-fat diet, which might be beneficial against hypertension [47]. Fatty fish and fish liver oil, egg yolk, or dairy products are the main dietary sources of vitamin D, but their consumption patterns vary according to age and eating habits [48], and this might be partly ascribed to the inverse association between dietary exposures to BCAAs and vitamin D.

It must be acknowledged that there are practical limitations of our study. First, the defects and various biases of a cross-sectional study only allowed us to explore the potential association between dietary BCAAs and physical function in the elderly population, but not to establish causality. Second, residual confounding could not be ignored in this observational study, despite our attempts to control for the major potential confounders by using several adjustment models. Third, the use of a self-reported FFQ might introduce measurement error and recall bias. However, a validated semiquantitative FFQ with pictures of foods in corresponding portion sizes has been adopted to minimize the risk. Finally, the generalizability of our findings to other populations might be limited because all of our participants were enrolled from Shenzhen, China.

## 5. Conclusions

This cross-sectional study suggested that increased dietary exposure to BCAAs was positively associated with improved muscle strength and better physical performance. Future research with a larger sample size is needed to further clarify the effect of each type of BCAA on physical performance and general well-being in elderly individuals.

## Figures and Tables

**Figure 1 nutrients-14-04367-f001:**
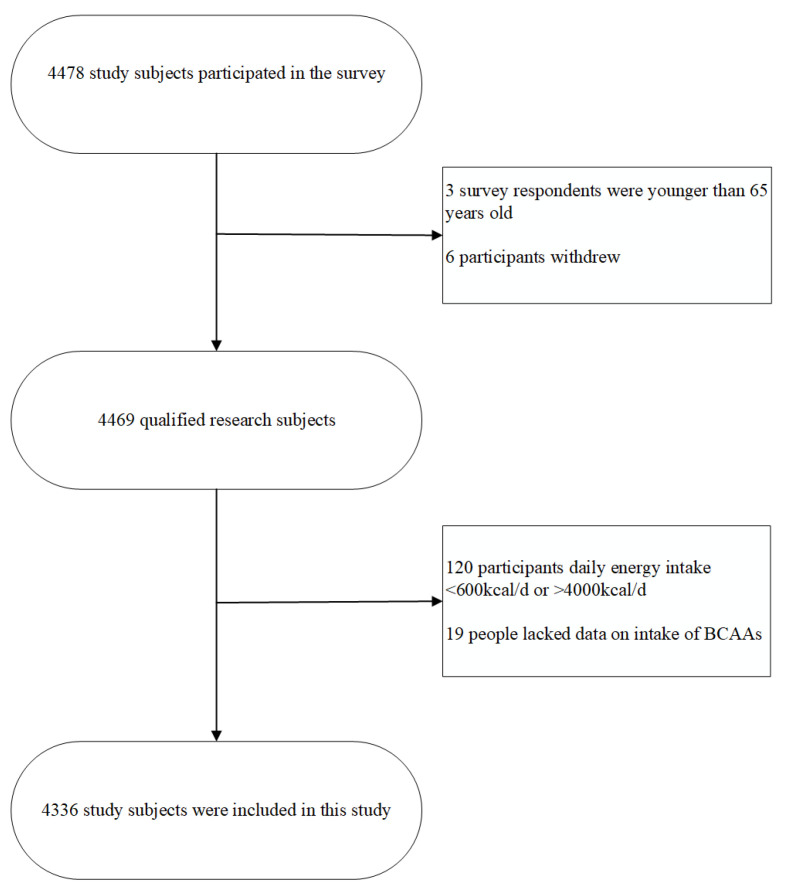
Flow chart.

**Table 1 nutrients-14-04367-t001:** Descriptive characteristics of study participants by quartiles of total BCAAs intake (*n* = 4336).

	Quartiles of BCAAs Intake ^a^					
	Q1 (*n* = 1084)	Q2 (*n* = 1084)	Q3 (*n* = 1084)	Q4 (*n* = 1084)	*p*-Value ^b^	*p*-Trend ^c^
Age (years), mean (sd)	72.49 (5.44)	72.77 (5.54)	73.07 (5.52)	72.57 (5.41)	0.070	0.475
Sex (male), *n* (%)	386.00 (35.6)	446.00 (41.1)	482.00 (44.5)	549.00 (50.6)	<0.001	<0.001
Household registration, *n* (%)					<0.001	<0.001
Large cities	500.00 (46.1)	591.00 (54.5)	748.00 (69.0)	753.00 (69.5)		
Small–medium cities	138.00 (12.7)	147.00 (13.6)	92.00 (8.5)	104.00 (9.6)		
Rural counties	446.00 (41.2)	346.00 (31.9)	244.00 (25.5)	227.00 (20.9)		
BMI (kg/m^2^)					0.199	0.598
<18.5	27.00 (27.84%)	20.00 (20.62%)	29.00 (29.90%)	21.00 (21.65%)		
18.5–23.9	430.00 (25.78%)	420.00 (25.18%)	403.00 (24.16%)	415.00 (24.88%)		
24.0–27.9	323.00 (21.96%)	375.00 (25.49%)	398.00 (27.06%)	375.00 (25.49%)		
≥28.0	107.00 (25.06%)	116.00 (27.17%)	96.00 (22.48%)	108.00 (25.29%)		
Smoking status, *n* (%)					0.154	0.058
Yes	200.00 (18.5)	232.00 (21.4)	222.00 (20.5)	241.00 (22.2)		
No	884.00 (81.5)	852.00 (78.6)	862.00 (79.5)	843.00 (77.8)		
Drinking status, *n* (%)					0.002	<0.001
Yes	137.00 (12.6)	148.00 (13.7)	175.00 (16.1)	195.00 (18.0)		
No	947.00 (87.4)	936.00 (86.3)	909.00 (83.9)	889.00 (82.0)		
Diabetes, *n* (%)	219.00 (20.2)	257.00 (23.7)	216.00 (19.9)	237.00 (21.9)	0.117	0.830
Hypertension, *n* (%)	515.00 (47.5)	508.00 (46.9)	479.00 (44.2)	465.00 (42.9)	0.099	0.015
Dyslipidemia, *n* (%)	158.00 (14.6)	168.00 (15.5)	146.00 (13.5)	158.00 (14.6)	0.614	0.672
GLU (mmol/L) mean (sd)	5.66 (1.64)	5.66 (1.63)	5.55 (1.54)	5.65 (1.6)	0.339	0.517
TC (mmol/L) mean (sd)	5.12 (1.26)	5.08 (1.12)	5.08 (1.14)	5.07 (1.52)	0.819	0.404
TG (mmol/L) mean (sd)	1.54 (1.23)	1.53 (0.9)	1.51 (0.93)	1.45 (1.1)	0.242	0.056
HDL-C (mmol/L) mean (sd)	1.55 (3.94)	1.38 (0.45)	1.4 (0.51)	1.39 (0.42)	0.117	0.207
LDL-C (mmol/L) mean (sd)	3.23 (0.89)	3.21 (0.92)	3.2 (0.88)	3.17 (0.85)	0.143	0.539
Handgrip strength, mean (sd)	21.33 (7.87)	22.77 (8.66)	23.50 (8.56)	24.31 (8.25)	<0.001	<0.001
4-m usual walking speed (m/s), mean (sd)	3.99 (1.36)	4.01 (1.78)	4.03 (4.49)	3.75 (1.60)	0.054	0.055
4-m fast walking speed (m/s), mean (sd)	3.01 (0.95)	2.99 (1.33)	2.87 (0.88)	2.77 (1.05)	<0.001	<0.001
Repeated chair rises (s), mean (sd)	11.74 (3.93)	11.43 (3.82)	11.11 (3.68)	10.84 (3.75)	<0.001	<0.001

Abbreviations: BCAAs, Branched-chain amino acids; Q, quartile; sd, standard deviation; BMI, body mass index; GLU, fasting blood glucose; TC, total cholesterol; TG, total triglycerides; HDL-C, high density lipoprotein; LDL-C, ligh density lipoprotein. Note: ^a^ Cutoff values of BCAA quartiles are as follows: Q1: <13,210.90 mg/day, Q2: 13,210.90~18,289.47 mg/day, Q3: 18,289.47~24,360.62 mg/day, Q4: ≥24,360.62 mg/day; ^b^ *p*-value was calculated using the ANCOVA analysis for difference across quartiles of each type of BCAAs; ^c^ *p*-trend was determined using a test for linear trend across quartiles of BCAAs.

**Table 2 nutrients-14-04367-t002:** Leading seven nutrients of study participants by quartiles of dietary BCAAs (*n* = 4336).

	Quartiles of BCAA Intake ^a^				*p*-Value ^b^	*p*-Trend ^c^
	Q1	Q2	Q3	Q4		
Energy intake (kcal/day)	1159.28 ± 446.25	1341.52 ± 394.68	1477.41 ± 389.17	1702.02 ± 561.19	<0.001	<0.001
Fat (g/day)	129.84 ± 118.67	117.00 ± 89.61	114.89 ± 100.02	117.49 ± 92.65	0.001	0.004
Protein (g/day)	53.14 ± 14.46	54.35 ± 15.68	53.96 ± 14.01	54.14 ± 15.9	0.261	0.201
Carbohydrate (g/day)	104.23 ± 91.13	112.08 ± 90.9	117.92 ± 89.99	113.86 ± 91.10	0.004	0.005
Dietary soluble fiber (g/day)	6.46 ± 8.44	6.74 ± 7.83	7.09 ± 7.93	6.66 ± 6.93	0.297	0.377
Vitamin D (µg/day)	147.25 ± 213.07	123.17 ± 157.57	121.84 ± 202.51	128.58 ± 169.44	0.005	0.025
Folate (µg/day)	173.85 ± 131.18	177.56 ± 120.1	192.11 ± 130.96	185.63 ± 132.82	0.005	0.005

Abbreviations: BCAAs, Branched-chain amino acids; Q, quartile. Note: Data are presented as mean ± standard deviation (SD). ^a^ Cutoff values of BCAA quartiles are as follows: Q1: <13,210.90 mg/day, Q2: 13,210.90~182,89.47 mg/day, Q3: 18,289.47~24,360.62 mg/day, Q4: ≥24,360.62 mg/day. ^b^ *p*-value was calculated using the ANCOVA analysis for difference across quartiles of each type of BCAAs; ^c^ *p*-trend was determined using a test for linear trend across quartiles of BCAAs.

**Table 3 nutrients-14-04367-t003:** Covariate-adjusted mean changes in four physical performance indicators by quartiles of total BCAAs intakes (*n* = 4336).

	Quartiles of BCAAs Intake ^a^									
	Q1		Q2		Q3		Q4		MD ^b^	*p*-Trend ^c^
	Mean	SE	Mean	SE	Mean	SE	Mean	SE		
Handgrip strength (kg)										
Crude	21.4	0.25	22.8	0.25	23.6	0.25	24.4	0.25	2.97	<0.001
Model 1	22.6	0.22	23.6	0.21	24.2	0.22	24.3	0.21	1.65	<0.001
Model 2	22.6	0.34	23.5	0.33	24.1	0.33	24.3	0.33	1.64	<0.001
Model 3	21.4	0.34	23.4	0.33	24.0	0.33	24.4	0.33	1.74	<0.001
4-m usual walking speed (m/s)										
Crude	4.1	0.08	4.1	0.08	4.1	0.08	3.8	0.08	−0.29	0.034
Model 1	4.1	0.08	4.1	0.08	4.0	0.08	3.8	0.08	−0.25	0.086
Model 2	4.1	0.11	4.1	0.11	4.2	0.11	3.9	0.11	−0.23	0.128
Model 3	4.1	0.12	4.1	0.11	4.2	0.11	3.9	0.11	−0.22	0.122
4-m fast walking speed (m/s)										
Crude	3.1	0.03	3.0	0.03	2.9	0.03	2.8	0.03	−0.28	<0.001
Model 1	3.1	0.03	3.0	0.03	2.9	0.03	2.8	0.03	−0.24	<0.001
Model 2	3.1	0.05	3.0	0.05	2.9	0.05	2.9	0.05	−0.25	<0.001
Model 3	3.1	0.05	3.0	0.05	2.9	0.05	2.9	0.04	−0.26	<0.001
Repeated chair rises (s)										
Crude	11.8	0.12	11.5	0.12	11.1	0.12	10.5	0.12	−0.93	<0.001
Model 1	11.8	0.12	11.5	0.12	11.1	0.12	10.9	0.71	−0.83	<0.001
Model 2	12.0	0.15	11.5	0.15	11.3	0.15	11.1	0.15	−1.11	<0.001
Model 3	12.0	0.15	11.5	0.15	11.3	0.15	11.1	0.15	−0.90	<0.001

Abbreviations: BCAAs, Branched-chain amino acids; Q, quartile; SE, standard error; MD, mean difference. Note: ^a^ Cutoff values of BCAA quartiles are as follows: Q1: <13,210.90 mg/day, Q2: 13,210.90~182,89.47 mg/day, Q3: 18,289.47~24,360.62 mg/day, Q4: ≥24,360.62 mg/day; ^b^ Mean difference between quartile 4 and quartile 1 was calculated by ANCOVA. ^c^ *p*-trend was determined using a test for linear trend across quartiles of BCAAs. Model 1 was adjusted for age, sex. Model 2 was additionally adjusted for BMI, smoking status, diabetes, hypertension, drinking status. Model 3 was additionally adjusted for vitamin D, fat, carbohydrate.

**Table 4 nutrients-14-04367-t004:** Odds ratios of weak muscle strength or decline in physical function by quartiles of BCAAs intake.

	Quartiles of BCAAs Intake ^a^			
	Q1	Q2	Q3	Q4
Weak muscle strength				
Hang grip (<18 kg for female)				
Crude	1 (Ref)	0.77 (0.62–0.96) *	0.67 (0.54–0.84) *	0.55 (0.43–0.69) *
Model 1	1 (Ref)	0.74 (0.59–0.93) *	0.64 (0.51–0.81) *	0.54 (0.42–0.68) *
Model 2	1 (Ref)	0.73 (0.57–0.94) *	0.64 (0.50–0.82) *	0.51 (0.39–0.67) *
Model 3	1 (Ref)	0.72 (0.56–0.92) *	0.62 (0.48–0.80) *	0.50 (0.38–0.65) *
Hang grip (<28 kg for male)				
Crude	1 (Ref)	0.68 (0.52–0.90) *	0.80 (0.61–1.05)	0.65 (0.50–0.85) *
Model 1	1 (Ref)	0.67 (0.50–0.89) *	0.77 (0.58–1.02)	0.65 (0.49–0.85) *
Model 2	1 (Ref)	0.68 (0.50–0.92) *	0.80 (0.59–1.08)	0.68 (0.51–0.92) *
Model 3	1 (Ref)	0.67 (0.49–0.91) *	0.79 (0.58–1.07)	0.67 (0.50–0.91) *
Physical performance decline				
Slow 4-m usual walking speed (<0.8 m/s)				
Crude	1 (Ref)	0.90 (0.71–1.15)	0.83 (0.65–1.06)	0.60 (0.46–0.79) *
Model 1	1 (Ref)	0.88 (0.68–1.14)	0.80 (0.62–1.04)	0.62 (0.47–0.82) *
Model 2	1 (Ref)	0.93 (0.70–1.23)	0.90 (0.68–1.19)	0.68 (0.50–0.92) *
Model 3	1 (Ref)	0.93 (0.70–1.24)	0.91 (0.68–1.21)	0.68 (0.50–0.93) *
Slow repeated chair rises (≥12 m/s)				
Crude	1 (Ref)	0.86 (0.72–1.03)	0.79 (0.66–0.95) *	0.65 (0.54–0.78) *
Model 1	1 (Ref)	0.86 (0.71–1.03)	0.77 (0.64–0.93) *	0.66 (0.55–0.80) *
Model 2	1 (Ref)	0.79 (0.65–0.97) *	0.77 (0.63–0.94) *	0.66 (0.54–0.81) *
Model 3	1 (Ref)	0.79 (0.64–0.96) *	0.77 (0.63–0.94) *	0.66 (0.54–0.81) *

Abbreviations: BCAAs, Branched-chain amino acids; Q, quartile. Note: ^a^ Cutoff values of BCAA quartiles are as follows: Q1: <13,210.90 mg/day, Q2: 13,210.90~182,89.47 mg/day, Q3: 18,289.47~24,360.62 mg/day, Q4: ≥24,360.62 mg/day; * *p* < 0.05. Model 1 was adjusted for age and sex. Model 2 was additionally adjusted for BMI, smoking status, diabetes, hypertension, and drinking status. Model 3 was additionally adjusted for vitamin D, fat, and carbohydrates.

**Table 5 nutrients-14-04367-t005:** Covariate-adjusted mean changes in four physical performance indicators by quartiles of isoleucine, leucine, valine intakes (*n* = 4336).

	Isoleucine					Leucine					Valine				
	Q1 ^a^		Q4 ^a^		*p*-Trend ^b^	Q1 ^a^		Q4 ^a^		*p*-Trend ^b^	Q1 ^a^		Q4 ^a^		*p*-Trend ^b^
	Mean	SE	Mean	SE		Mean	SE	Mean	SE		Mean	SE	Mean	SE	
Handgrip strength (kg)															
Crude	23.1	0.26	23.2	0.26	0.784	23.0	0.26	23.0	0.26	0.938	23.0	0.26	23.0	0.26	0.938
Model 1	23.7	0.22	23.7	0.22	0.978	23.7	0.22	23.7	0.22	0.999	23.6	0.22	23.6	0.22	0.927
Model 2	23.8	0.29	23.7	0.28	0.781	23.8	0.29	23.7	0.28	0.816	23.7	0.29	23.6	0.28	0.709
Model 3	23.7	0.29	23.7	0.28	0.927	23.7	0.29	23.7	0.28	0.548	23.6	0.29	23.6	0.28	0.596
4-m usual walking speed															
Crude	4.0	0.08	3.9	0.08	0.701	4.0	0.08	4.0	0.08	0.762	4.0	0.08	4.0	0.08	0.774
Model 1	4.0	0.08	4.0	0.08	0.836	4.0	0.08	4.0	0.08	0.814	4.0	0.08	4.0	0.08	0.870
Model 2	4.1	0.11	4.0	0.11	0.885	4.1	0.11	4.0	0.11	0.859	4.0	0.11	4.0	0.11	0.923
Model 3	4.0	0.11	4.0	0.11	0.985	4.0	0.11	4.0	0.11	0.949	4.0	0.11	4.0	0.11	0.996
4-m fast walking speed															
Crude	3.0	0.03	2.9	0.03	0.308	3.0	0.03	2.9	0.03	0.397	3.0	0.03	2.9	0.03	0.538
Model 1	3.0	0.03	2.9	0.03	0.429	3.0	0.03	2.9	0.03	0.537	3.0	0.03	2.9	0.03	0.646
Model 2	3.0	0.04	2.9	0.04	0.540	3.0	0.04	2.9	0.04	0.654	2.9	0.04	2.9	0.04	0.766
Model 3	3.0	0.04	2.9	0.04	0.544	3.0	0.04	2.9	0.04	0.664	2.9	0.04	2.9	0.04	0.775
Repeated chair rises															
Crude	11.4	0.12	11.1	0.12	0.259	11.4	0.12	11.2	0.12	0.141	11.4	0.12	11.2	0.12	0.134
Model 1	11.4	0.12	11.2	0.12	0.117	11.4	0.12	11.2	0.12	0.165	11.4	0.12	11.2	0.12	0.184
Model 2	11.5	0.15	11.4	0.15	0.312	11.5	0.15	11.4	0.40	0.852	11.5	0.15	11.4	0.15	0.757
Model 3	11.5	0.15	11.4	0.15	0.473	11.5	0.15	11.4	0.15	0.327	11.5	0.15	11.4	0.15	0.645

Abbreviations: BCAAs, Branched-chain amino acids; Q, quartile; SE, standard error. Note: ^a^ Cutoff values of BCAA quartiles are as follows: Isoleucine: Q1: <659.69 mg/day, Q2: 659.69~850.09 mg/day, Q3: 850.09~934.89 mg/day, Q4: ≥934.89 mg/day; Leucine: Q1: <1268.53 mg/day, Q2: 1268.53~1630.08 mg/day, Q3: 1630.08~1781.98 mg/day, Q4: ≥1781.98 mg/day; Valine: Q1: <876.02 mg/day, Q2: 876.02~1158.71 mg/day, Q3: 1158.71~1257.01 mg/day, Q4: ≥1257.01 mg/day; ^b^ *p*-trend was determined using a test for linear trend across quartiles of BCAAs. Model 1 was adjusted for age and sex. Model 2 was additionally adjusted for BMI, smoking status, diabetes, hypertension, and drinking status. Model 3 was additionally adjusted for vitamin D, fat, and carbohydrates.

## Data Availability

The data presented in this study are available on request from the corresponding author.

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
