# Peer review of "Association between Branched-Chain Amino Acid Intake and Physical Function among Chinese Community-Dwelling Elderly Residents"

_nutrients, 2022, doi:10.3390/nu14204367_

Round 1

Reviewer 1 Report

Dear authors,

First, I would like to congratulate you for the great article produced by you, however, I would like to ask you some questions, such as:

The cardiovascular parameters of the study participants?;

If yes, did these parameters undergo significant or significant changes?;

The reason for my questions is that I have observed that the literature data are conflicting regarding the effects promoted by branched-chain amino acid supplementation on cardiac activity and in patients with heart failure (Int Heart J . 2021 Nov 30;62(6):1342-1347. doi: 10.1536/ihj.21-102; Theranostics . 2020 Apr 27;10(12):5623-5640. doi: 10.7150/thno.44836; circulation . 2016 May 24;133(21):2038-49. doi: 10.1161/CIRCULATIONAHA.115.020226) and, therefore, I understand that it would be interesting and enriching for you to discuss in more detail the cardiovascular parameters of the study participants, given that these are Chinese adult patients aged 65 years and over.

Once again I would like to commend the great work done by you.

Kind regards,

Author Response

Authors Response on Reviewer's Comments

1. First, I would like to congratulate you for the great article produced by you, however, I would like to ask you some questions, such as: The cardiovascular parameters of the study participants?;

If yes, did these parameters undergo significant or significant changes?;

Response: Thank you so much for your review and suggestions. We have added the related cardiovascular parameters (fasting glucose, HDL, LDL, TC, TG) in the baseline table (table 1) but we did not find any significant difference of these parameters across quartiles of BCAAs. Given this is a cross-sectional study based on the baseline data from the Nanshan elderly cohort study, we did not have the secular data for cardiovascular parameters and we could not observe any change of these parameters.

“2.2 Covariate collection

Potential confounders such as sociodemographic factors (age, sex, household registration, body mass index [BMI, kg/m2], etc.), health-related behaviors (smoking status, alcohol consumption, and physical activities), medical histories (diabetes, hypertension, dyslipidemia, etc.), and family histories of diseases were collected by trained investigators with relevant medical knowledge through face-to-face interviews using a structured questionnaire. The latest laboratory data including fasting blood glucose (GLU), total cholesterol (TC), total triglycerides (TG), high- and low-density lipoprotein (HDL-C and LDL-C) were collected from electronic report in community health service centers. Weight and height were measured, and BMI was calculated as weight (kg)/height (m2).”

2. The reason for my questions is that I have observed that the literature data are conflicting regarding the effects promoted by branched-chain amino acid supplementation on cardiac activity and in patients with heart failure (Int Heart J . 2021 Nov 30;62(6):1342-1347. doi: 10.1536/ihj.21-102; Theranostics . 2020 Apr 27;10(12):5623-5640. doi: 10.7150/thno.44836; circulation . 2016 May 24;133(21):2038-49. doi: 10.1161/CIRCULATIONAHA.115.020226) and, therefore, I understand that it would be interesting and enriching for you to discuss in more detail the cardiovascular parameters of the study participants, given that these are Chinese adult patients aged 65 years and over.

Response: Thank you so much for your suggestion. As it shown in supplementary materials, we have conducted several sensitivity analyses excluding those who had been diagnosed with major cardiovascular diseases (coronary heart disease, myocardial infarction, stroke, and angina pectoris) to explore potential effect of these comorbidities on physical performance, but most of the results between total dietary exposure to BCAAs and physical performance remained robust in all four subpopulations results in these analyses (sTables 1-5). So, we did not mention the details of this part of results in discussion.

Reviewer 2 Report

Line 68, correct grammatical error.

Line 95, correct grammatical error.

Line 184, Is 24g/day a rather high intake? Highest group in a recent Nature study consumed around 19g/day.

Line 191, 192, 193; "Compared to those with the lowest BCAA intake (Q1),  those with the highest dietary intake of BCAAs (Q4) tended to have higher intake levels of energy, fat, and vitamin D". This appears to be inaccurate, Q4 had a lower intake of fat than Q1, not a higher intake.

Table 2- Total energy intake for Q1 is presumably a misprint? 115.28 kcal/d seems unfeasibly low?

Line 201, 202; The formatting of the numbers is incorrect, e.g., "132,10.90 mg/d". Should this be 13,210.90 mg/d?

Table 3- There are no units given for the four-meter walking and repeated chair rises data. Should it be seconds?

Line 232,233; Formatting of the numbers is incorrect (see earlier comment)

Line 261, 262; Formatting of the numbers is incorrect (see earlier comment)

Table 5- There are no units given for the four-meter walking and repeated chair rises data. Should it be seconds?

Line 316- correct grammatical error.

Line 357- correct grammatical error.

Line 361- correct grammatical error.
